# High Electrocaloric Effect in Lead Scandium Tantalate Thin Films with Interdigitated Electrodes

**DOI:** 10.3390/s22114049

**Published:** 2022-05-27

**Authors:** Veronika Kovacova, Sebastjan Glinsek, Stephanie Girod, Emmanuel Defay

**Affiliations:** Materials Research and Technology Department, Luxembourg Institute of Science and Technology (LIST), 41 Rue du Brill, L-4422 Belvaux, Luxembourg; sebastjan.glinsek@list.lu (S.G.); stephanie.girod@list.lu (S.G.); emmanuel.defay@list.lu (E.D.)

**Keywords:** electrocalorics, lead scandium tantalate, ferroelectric thin films, thin films deposition

## Abstract

Lead scandium tantalate, Pb(Sc,Ta)O_3_, is an excellent electrocaloric material showing large temperature variations, good efficiency, and a broad operating temperature window. In form of multilayer ceramic capacitors integrated into a cooling device, the device can generate a temperature difference larger than 13 K. Here, we investigate Pb(Sc,Ta)O_3_ in form of thin films prepared using the sol–gel chemical solution deposition method. We report the detailed fabrication process of high-quality films on various substrates such as c-sapphire and fused silica. The main originality of this research is the use of interdigitated top electrodes, enabling the application of very large electric fields in PST. We provide structural and electrical characterisation, as well as electrocaloric temperature variation, using the Maxwell relation approach. Films do not show a B-site ordering. The temperature variation from 7.2 to 15.7 K was measured on the Pb(Sc,Ta)O_3_ film on a c-sapphire substrate under the electric field of 1330 kV/cm between 14.5 °C and 50 °C. This temperature variation is the highest reported so far in Pb(Sc,Ta)O_3_ thin films. Moreover, stress seems to have an effect on the maximum permittivity temperature and thus electrocaloric temperature variation with temperature in Pb(Sc,Ta)O_3_ films. Tensile stress induced by fused silica shifts the “transition” of Pb(Sc,Ta)O_3_ to lower temperatures. This study shows the possibility for electrocaloric temperature variation tuning with stress conditions.

## 1. Introduction

In recent years, lead scandium tantalate ceramics, PbSc_0.5_Ta_0.5_O_3_ (PST), have drawn considerable attention, due to their excellent electrocaloric properties near room temperature. In form of bulk ceramics, PST exhibits an isothermal entropy change *Δ**S* of 2.6 J/(K.kg) and an adiabatic temperature change Δ*T* of 2.3 K under 26 kV/cm [1,2]. In terms of efficiency 𝜂, the bulk PST is exceptionally efficient with 𝜂 larger than 80 in a narrow temperature range of 2.5 K [3]. When looking at its multilayer ceramic capacitor (MLCC) counterpart, the figures are even more impressive. PST layers are thinner in MLCC than in bulk form, which enables the application of larger electric fields. According to Nair et al., PST MLCC, when driven supercritically, shows an adiabatic Δ*T* of 5.5 K [4]. One of the key advantages of the MLCC form is its expanded working temperature window. Indeed, the PST exhibits a Δ*T* larger than 3 K in a temperature range of more than 170 K, when driven with 290 kV/cm. This large temperature range comes at the expense of PST MLCC efficiency, which is equal to 31 [5]. The PST MLCCs couple high adiabatic Δ*T* and a large working temperature range and are, therefore, excellent candidates for use in prototypes where regeneration plays a paramount role. In 2020, Torello et al. reported an electrocaloric cooling prototype showing a temperature span of 13 K [6].

These achievements have been possible due to improved ordering, *S_B_,* of PST. PST crystallises in the perovskite structure ABO_3_. Considering a pseudo-cubic structure, A-site atoms are situated in the corners, B-site atoms in the cube centre, and oxygens are in the middle of the cube’s faces. The order in PST refers to the chemical arrangement of the B-site atoms with regard to the neighbouring cubic unit cells in space. A fully ordered structure with *S* equal to 1 consists of (111) crystallographic planes, with scandium on the B-site alternating with (111) planes of tantalum on the B-site. This arrangement gives rise to a PST superstructure consisting of eight unit cells and extra diffraction peaks corresponding to twice the dimensions of the primary unit cell. In practice, the material is rarely fully ordered. The order parameter is determined as a ratio between integrated intensities of peaks in the X-ray diffraction (XRD) pattern corresponding to the PST structure and the PST superstructure, relative to the ratio of the same peaks in the case of the fully ordered pattern [7]. In other words, the order parameter corresponds to the number of superstructure-ordered regions within a disordered matrix. The presence of small, ordered clusters gives rise to the frequency dependence of the relative permittivity in PST and its relaxor behaviour [8,9]. It has been demonstrated that order and adiabatic Δ*T* are directly correlated. The higher the order, the higher the Δ*T* value [10]. Higher PST order is also accompanied by a stabilisation of relative permittivity values at around 8000 for ceramics with lower frequency dependency and a stabilisation of the ferroelectric phase [11]. Higher order leads to more latent heat generation and, thus, more first-order-like phase transition in PST near room temperature.

One can ask whether there is more room for improvement of PST. A very adapted form for studying PST material is thin films. Indeed, thin films enable the examination of the effects of factors such as strain and texture on the order parameter [12], application of higher electric fields, and effects of doping. The sol–gel deposition method is appealing due to its adaptable chemistry, lower cost in comparison to vacuum deposition techniques, and scalability. This technique has been adapted for the fabrication of PST thin films by [13,14,15,16,17,18,19]. Patel et al. studied the pyroelectric effect in PST films for thermal detector application and reported a comparable pyroelectric figure of merit than in PST ceramics. Fuflying et al. worked on the influence of strain on dielectric properties and found a 50% increase in permittivity upon the release of thin films from a silicon substrate. Huang et al. compared the microstructural and structural quality of PST thin films deposited with sol–gel, sputtering, and liquid delivery chemical vapour deposition. They reported dense microstructure and perovskite crystallographic structure in sol–gel-based and sputtered films. Liu et al., de Kroon et al., and Brinkman et al. studied processing and its effect on dielectric, pyroelectric, and piezoelectric properties. Only one article reports electrocaloric effects in PST thin films. Correia et al. studied the effect of the order on the frequency-dependent maximum permittivity temperature, *T*_m_, and estimated indirectly the change in entropy Δ*S* and Δ*T* of the electrocaloric effect with the Maxwell relation. In the partially ordered (*S_B_* = 0.32) PST film, deposited on a platinized silicon substrate, the estimated Δ*T* was as high as −6.9 K when 774 kV/cm was applied between 100 and 120 °C [19].

Correia et al. used the classic metal–insulator–metal (MIM) geometry which enables electrical probing in the out-of-plane direction with a maximum electric field of 774 kV/cm. On the other hand, the interdigitated electrode (IDE) geometry with the SU8 epoxy encapsulating layer enables the application of electric fields in the in-plane direction up to 4 MV/cm, as stated in [20]. This is interesting because the entropy change and the electrocaloric effect temperature variation (Equations (1) and (2) below) are proportional to the electric field. In addition to a higher electrical breakdown strength, the advantage of IDE is the possibility of longer electrical cycling. It is worth noting that Brinkman et al. [12] already used MIM and IDE geometry on PST films on c-sapphire for relative permittivity studies. Results of his research show that there is no film thickness dependency of permittivity in IDE geometry, the relative permittivity values from MIM and IDE electrodes are identical, and there is no “passive” layer effect in the IDE geometry. The electric field distribution in IDE geometry was modelled in Nguyen et al. [21], proposing equations for precise estimation of permittivity from capacitance values.

This manuscript reports large electrocaloric effects in sol–gel PST thin films. 160 nm thick films were deposited on c-sapphire and on fused silica substrates (FS). Samples’ crystallographic structure was characterised using XRD. Electrodes were patterned, and the samples were electrically tested. The main originality of this paper is the use of the IDE geometry with SU8 encapsulation layer enabling the application of very high electric fields up to 1330 kV/cm. The maximum electrocaloric response of 15.7 K on c-sapphire and 14.0 K on fused silica was calculated from the indirect approach in a temperature range between 14.5 °C and 50 °C.

## 2. Materials and Methods

Solutions were processed using Ta(V) ethoxide (99.98%, Merck, New York, NY, USA), Sc(III) acetate hydrate (99.9%, Merck), Pb(II) acetate trihydrate (99.5%, Merck) as metal precursors, and 2-methoxyethanol (99.3%, Merck) as a solvent. Sc and Pb precursors were freeze-dried prior to use to remove hydrated water.

Ta and Sc precursors and the solvent were weighted in a glove box and were refluxed for 24 h in an argon atmosphere. After cooling to room temperature, Pb precursor was added (30% excess), followed by an additional 2 h of reflux and final distillation. The final solution had a molarity of 0.3 M.

The prepared solution was deposited on c-sapphire (Siegert Wafer, Aachen, Germany) and on fused silica (Siegert Wafer). Prior to the deposition of PST sol, samples were coated with one layer of PbTiO_3_ in order to initiate the perovskite structure and enhance (100) orientation [12,22]. After this step, PST sol was spun on the samples at 3000 rpm, dried at 130 °C, and pyrolysed at 350 °C on a hot plate. After 4 deposited layers, the samples were annealed at 900 °C in a rapid thermal annealing furnace (air atmosphere, 15 min, heating rate of 50 °C/s). These two samples are discussed in the main paper. Additional platinised silicon, c-sapphire, and FS samples were prepared and annealed at 750 °C (see Appendix A for samples summary). For clarity of the main paper, results from additional samples are presented in Appendix A. PST thickness on all samples is 160 nm. It is worth noting that the synthesis method and deposition conditions were adapted from [12,18].

Lithography and lift-off processes were used to pattern the electrodes. The LOR3A and S1813 lithography resists were spun on samples and baked. An IDE pattern with a 3 μm wide gap, 50 pairs of 5 μm wide, and 370 μm long fingers, was used for c-sapphire and FS samples (Figure 1). The patterns were drawn in resists using a maskless aligner MLA 150 tool from Heidelberg Instruments, with 375 nm laser wavelength. The electrode surface area was calculated using the approach in [21].

Briefly, 100 nm of platinum was deposited using an in-house DC sputtering prototype under an argon atmosphere, at 2.1 × 10^−2^ mbar of pressure, and plasma was generated with the power of 200 W. The deposition time was 3 min and 40 s. The Pt electrodes were post-annealed at 400 °C for 5 min.

Then, a 2 μm thick SU8 resist was spun and patterned on the IDE structure in order to enable high-field application in PST films [20].

XRD patterns were collected using a D8 diffractometer (Bruker, Billerica, MA, USA) in reflexion geometry. The 2θ range was from 10° to 65° for all samples. The step size was 0.02° with 2 s of time per step. In order to index diffraction peaks, the pseudo-cubic space group Fm-3m from the PDF file number 01-074-2635 was used [23]. To estimate the order parameter S_B_ in thin films, one has to take into account that thin films are not randomly oriented, as indicated in [7]. Therefore, an adapted approach was considered. We compared the peak area of (111) planes, corresponding to the ordered PST superstructure, and the peak area corresponding to the (222) planes, present regardless of the order. *S_B_* factor was calculated as follows:$$S_{B\;111}^{2}=\frac{\left(\frac{I111}{I222}\right)film}{\left(\frac{I111}{I222}\right)ordered}$$
where *I*_111_/*I*_222_ for ordered PST is equal to 1.309, as calculated from [7,23]. A detailed explanation is given in Appendix A. In order to ensure the highest intensity of peaks of interest, we collected (111) and (222) peaks corresponding to the main texture and, thus, out of the plane. The areas of interest—namely, between 17° and 20° in 2θ corresponding to the location of (111) planes and between 37° and 39° corresponding to the (222) planes—were collected for 20 s per point with a 1 mm collimated beam.

Electrical characterisation was performed using an Aixacct system TFA2000E with FE-Module. Small field-capacitance measurements at 1 kHz with a 150 mV bias field were performed at temperatures from −10 to 50 °C. Polarisation measurements versus voltage up to 400 V at 100 Hz were performed at temperatures from 10 to 50 °C.

The electrocaloric effect magnitude was calculated using the indirect approach stemming from the Maxwell relations as detailed in [24]. From polarisation versus electric field, measurements were extracted maximum polarisation versus temperature values at a constant field. The derivative of polarisation versus temperature was then used to estimate the isothermal entropy change Δ*S* and the adiabatic temperature variation Δ*T*, as expressed in Equations (1) and (2):(1)$$\Delta S=\int_{E_{1}}^{E_{2}}{\left(\frac{\partial D_{i}}{\partial T}\right)}_{X,E}dE\;$$
(2)$$\Delta T=-\frac{1}{\rho }\int_{E_{1}}^{E_{2}}\frac{T}{C}{\left(\frac{\partial D_{i}}{\partial T}\right)}_{X,E}dE$$
where *ρ* stands for density; *T* stands for temperature; *E*_1_ and *E*_2_ for starting and final electric field; *C* stands for specific heat; *D* stands for the electric displacement and *X* for stress. It is worth noting that *D* can be approximated with polarisation *P* in the case of materials with a large dielectric constant, which is the case of PST. The value of specific heat *C* taken is 300 J kg^−1^ K^−1^ and density *ρ* is 9070 kg m^−3^ [3].

## 3. Results and Discussion

The films show a large columnar grain structure (~200 nm and more), with porosity in the cross-sectional micrograph (see Appendix A). No pyrochlore phase is observed on the micrographs.

Figure 1 displays the IDE geometry used for PST deposited on c-sapphire and fused silica. Figure 2 shows the XRD patterns of the two samples. The XRD data show that PST thin films crystallise in the perovskite phase without pyrochlore. Samples have two textures—namely, (200) and (220). Considering the PDF number 01-074-2635, the (200) peak intensity of randomly oriented PST powder is 64, whereas the intensity of (220) is 1000. The (220) peak outweighs the (200) peak by more than a factor of 15. The peaks (220) and (200) were fitted with Pymca software and area pseudo-Voigt function with linear function for background. The ratio of integrated areas of (220) and (200) is equal to 2.21 and 0.53 for PST on c-sapphire and FS, respectively. Thus, the main texture on all PST samples regardless of substrate and annealing temperature is (200) (see Appendix A for XRD patterns of all samples). In order to determine the order parameter of PST films, (111) and (222) peaks corresponding to the (200) texture were collected out of the plane, at χ = 54.7°. Details about order determination are in Appendix A.

The PST films on c-sapphire and FS do not show any (111) peak at 18.87°; therefore, they are not ordered. The only sample that shows partial ordering is PST on platinised silicon, with *S_B_* equal to 0.17 (Appendix A). The order parameter reported by Correia et al. in PST on platinised silicon was 0.32 [19], although the manuscript provides very few details about the order determination, and it remains unclear whether the author assumed random orientation of the PST film for order calculations. Brinkman et al. reported the appearance of significant order (*S_B_* = 0.22) in PST on platinised silicon upon annealing at 800 °C for a time as short as 20 min [12]. Notably, longer annealing times led to the degradation of the platinum bottom electrode. On the other hand, the FS and c-sapphire substrates are better suited for high temperatures and longer annealing times. Brinkman achieved no order in PST on c-sapphire annealed at 850 °C for 1 min. Significant order of 0.55 was achieved when PST film on c-sapphire was annealed for 1 h at 850 °C. An S equal to 0.91 was obtained for an annealing time as long as 35 h. Annealing at temperatures higher than 1000 °C results in PST film degradation. In our case, we consider that there is no ordering in our samples on c-sapphire and FS substrates.

Figure 3 shows relative permittivity values at zero fields for PST samples as a function of temperature. The relative permittivity of PST films deposited on c-sapphire peaks at 3000. The relative permittivity of PST films deposited on FS peaks at 850. There is a factor of 3 in *ε**_r_* of PST in c-sapphire and on FS. Similar values were reported by Brinkman [12]. Indeed, a zero ordered PST on c-sapphire (annealed at 850 °C for 1 min) showed relative permittivity as large as 3000. Relative permittivity values found for PST films are in the order of magnitude corresponding to disordered PST thin films. Brinkman reported that the relative permittivity of ordered PST thin films is around 6000, and permittivity for disordered films is down to 3000 [12].

It is worth noting that the maximum *ε**_r_* is at around 25 °C for samples on c-sapphire, whereas it is −3 °C for samples on FS substrates. *Tm* is defined as the temperature at maximum *ε**_r_*. Therefore, these results are similar to the findings of Brinkman et al. [18]. It is reported that the *Tm* for PST films under tensile or compressive stresses is shifted to lower temperatures than *Tm* for unstrained PST films. PST on c-sapphire is slightly compressive (−20 MPa [12]), whereas PST on FS is under large tensile stresses (+752 MPa see Appendix A [25]). This tendency is confirmed in all of the samples (see Appendix A). Thus, these results show that *Tm* is at around 20 °C for c-sapphire, and it is shifted to lower temperatures for PST on fused silica. The dielectric loss is gradually decreasing with increasing temperature for PST on fused silica and c-sapphire.

Figure 4 shows polarisation versus electric field at different temperatures. Samples on c-sapphire and on fused silica were able to withstand voltages up to 400 V corresponding to an electric field of 1330 kV/cm. All samples with IDE geometry withstand 400 V. This is the highest electric field reported for PST thin films. The IDE geometry enables a real innovation in terms of sample breakdown. Electrical characterisation results on additional samples are provided in Appendix A.

Figure 5 shows the maximum polarisation versus temperature for 1330 kV/cm for c-sapphire and FS.

One can see that the maximum polarisation decreases with increasing temperature regardless of the sample substrate, as shown in Figure 5. The decreasing trend of maximum polarisation indicates a negative value of the derivative $\frac{dP}{dT}$, which gives in return a negative *Δ**S* value and a positive Δ*T* (cf. Equations (1) and (2)).

Figure 6 shows Δ*S* and Δ*T* as a function of temperature for 1330 kV/cm for PST on c-sapphire and fused silica from 12 °C to 48 °C, obtained from Equations (1) and (2) and the data from Figure 5. The Δ*S* is equal to 149 kJ K^−1^ m^−3^ for PST on c-sapphire at 14.5 °C, which is equivalent to 16 J K^−1^ kg^−1^. The Δ*S* is 132 kJ K^−1^ m^−3^ for PST on FS, which is equivalent to 14.6 J K^−1^ kg^−1^. Notably, Correia et al. obtained Δ*S* equal to 6.3 J K^−1^ kg^−1^ for PST film under 774 kV/cm at 70 °C.

The largest Δ*T* of 15.7 K is achieved in PST at 14.5 °C on c-sapphire. Maximum Δ*T* for PST on FS reaches 14.0 K at 14.5 °C. Notably, samples annealed at 750 °C show slightly lower Δ*T* values (see Appendix A). Maximum Δ*T* for PST on platinised silicon with metal–insulator–metal (MIM) electrode geometry is 1.9 K at 12 °C and corresponds to 250 kV/cm (see Appendix A).

The reported values of Δ*T* on c-sapphire and on FS are large, despite the lack of order in PST films. This is due to the possibility to apply large electric fields with the encapsulated IDE geometry. The electric field is highest reported on thin films. Correia et al. reported temperature variation as large as 6.9 K under 774 kV/cm between 100 and 120 °C [19]. We applied 1330 kV/cm, which is more than times 1.7 larger field. For comparison, calculations of Δ*T* under 776 kV/cm are reported in Appendix A. Under 776 kV/cm, the highest Δ*T* is equal to 11.5 K and 9.1 K for c-sapphire and FS, respectively. These values are 1.3 et 1.7 times larger than those reported by Correia et al.

It is worth noting that the maximum Δ*T* reported by Correia et al. occurs between 100 and 120 °C. This result is unexpected because it is far from *Tm*, which is stated to be at around 20 °C. Our large Δ*T* occurs at 14.5 °C, which is expected because it is closer to *Tm*. The reported value of Δ*T* at 15 °C in Correia et al. is 3.8 K.

Interestingly enough, for PST on c-sapphire and FS, respectively, we estimated a Δ*T* larger than 7 K and 4.5 K for a temperature window between 14.5 to 50 °C. This observation confirms the statement of Crossley et al. and Correia et al. that clamping broadens and extends the electrocaloric effect to higher temperatures [1,19]. The temperature window of measured Δ*T* for bulk PST is equal to 17 K, whereas the window for clamped PST is 70 K [1]. The temperature window is stated to be 130 K for PST thin films [1,19].

The relative permittivity measurements showed that Tm is shifted towards lower temperatures for PST on FS in comparison to PST on c-sapphire. Moreover, the Δ*T* values are larger at temperatures closer to *Tm*. It can be hypothesised that, in the temperature range from 14.5 °C to 50 °C, the Δ*T* of FS samples is lower than the Δ*T* of c-sapphire because PST on FS is further away from *Tm*. This result opens the possibility of tuning the electrocaloric temperature variation in PST with stress.

## 4. Conclusions

To conclude, in this manuscript, we reported outstanding values of the electrocaloric temperature variations in PST thin films deposited on c-sapphire and on FS substrates, determined with the Maxwell relation. PST on c-sapphire showed a variation of almost 16 K under 1330 kV/cm at 14.5 °C. PST on FS showed a variation of 14 K under the same conditions. Large temperature variations were extracted due to the IDE geometry which enables the application of high electric fields and the possibility of longer cycling.

Moreover, thermal stress seems to play a paramount role in *Tm*, the temperature of maximum permittivity, in PST films. The Δ*T* values were higher at temperatures close to *Tm*. Therefore, there is an opportunity for the fine-tuning of maximum Δ*T* in temperature using stress conditions.

## Figures and Tables

**Figure 1 sensors-22-04049-f001:**
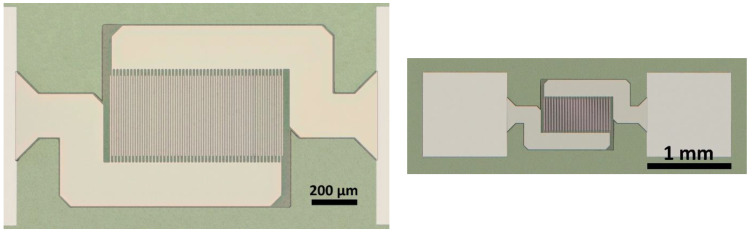
Images of the IDE electrode geometry.

**Figure 2 sensors-22-04049-f002:**
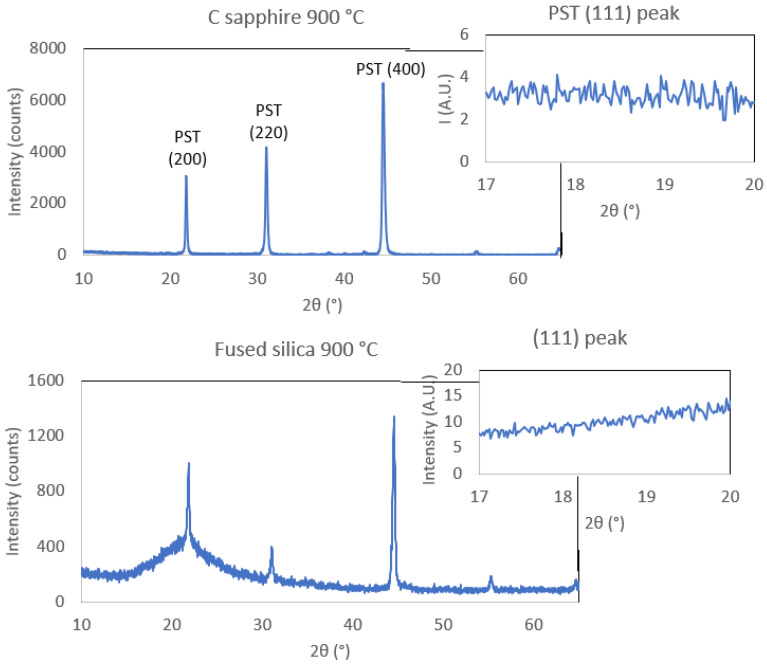
XRD patterns of PST samples with inset images of PST (111) peak located at χ = 54.7°.

**Figure 3 sensors-22-04049-f003:**
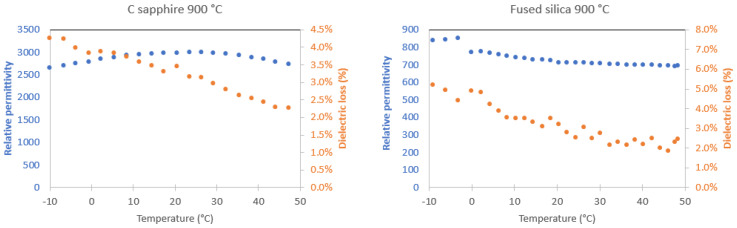
Permittivity versus temperature for PST on c-sapphire and FS substrates annealed at 900 °C. The measurements were performed on heating.

**Figure 4 sensors-22-04049-f004:**
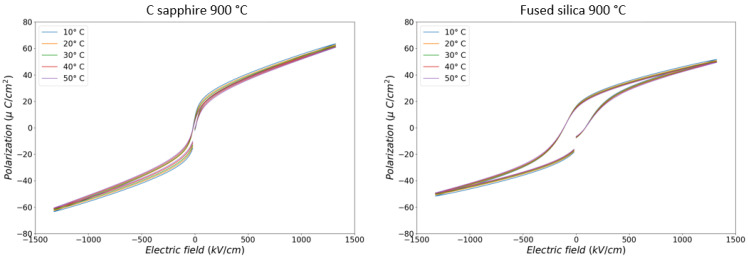
Polarisation vs. electric field loops for PST films.

**Figure 5 sensors-22-04049-f005:**
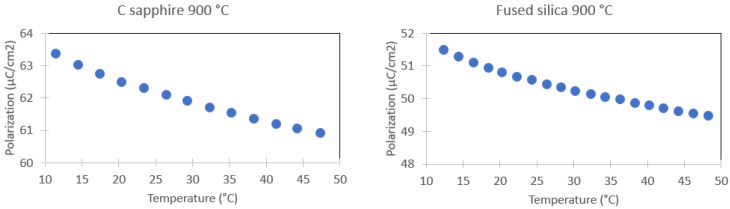
Maximum polarisation at 1330 kV/cm versus temperature for PST on c-sapphire and FS samples annealed at 900 °C. The measurements were performed on heating.

**Figure 6 sensors-22-04049-f006:**
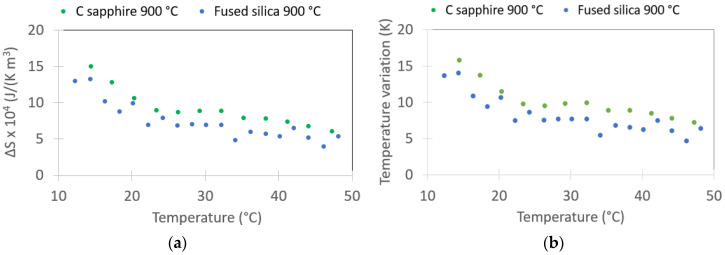
Calculated entropy variation (**a**) and calculated temperature variation (**b**) due to the electrocaloric effect vs. temperature. PST on fused silica and c-sapphire values correspond to 1330 kV/cm.

## Data Availability

Data is contained within the article or Appendix A.

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
