# Peer review of "High Electrocaloric Effect in Lead Scandium Tantalate Thin Films with Interdigitated Electrodes"

_sensors, 2022, doi:10.3390/s22114049_

Round 1
Reviewer 1 Report
Please refer to the attached file.

Author Response
Dear editor and reviewers,
First, we would like to thank you for your comments, suggestions, and questions. We have implemented changes in response to each point of view and we think that the manuscript has gained in scientific value as a result. The original review comments are shown in black. Answers to those comments are shown in blue and revisions to the manuscript are shown in red font.
Reviewer Comments:
Reviewer #1
Open Review
(x) I would not like to sign my review report
( ) I would like to sign my review report
English language and style
( ) Extensive editing of English language and style required
( ) Moderate English changes required
( ) English language and style are fine/minor spell check required
(x) I don't feel qualified to judge about the English language and style
|
Yes |
Can be improved |
Must be improved |
Not applicable |
|
|
Does the introduction provide sufficient background and include all relevant references? |
( ) |
(x) |
( ) |
( ) |
|
Are all the cited references relevant to the research? |
( ) |
(x) |
( ) |
( ) |
|
Is the research design appropriate? |
( ) |
(x) |
( ) |
( ) |
|
Are the methods adequately described? |
( ) |
(x) |
( ) |
( ) |
|
Are the results clearly presented? |
( ) |
(x) |
( ) |
( ) |
|
Are the conclusions supported by the results? |
( ) |
(x) |
( ) |
( ) |
The submitted work reports the electrocaloric temperature variations in PST thin films deposited on c-sapphire and on FS substrates. The microstructures of the PST thin films are characterized. Moreover, polarization and electrocaloric effect of the resultant films are proposed. Overall, the scientific quality of the submitted work is acceptable. The discussions based on the experimental results are sound. Several issues are proposed herein before it can be accepted for publication.
- Although Pb precursor (30 % excess) was used for synthesizing PST films, the composition of the PST films obtained from composition analysis measurements did not provide.
Lead is volatile and its often deficient when it comes to fabrication of thin films. Lead deficiency manifests as pyrochlore phase, present typically in Pb(Zr,Ti)O3, Pb(Mn,Nb)O3 [1], and in Pb(Sc,Ta)O3 [2] sol-gel thin films. In order to prevent Pb deficiency, Pb precursor is introduced in excess in the sol. The excess of Pb precursor in PST film fabrication is an approach that has been used in several articles studying processing of PST films with sol-gel method. Brinkman et al. compares PST samples with Pb excess varying from 0 to 30 mol% and shows less pyrochlore phase with increased Pb excess. We introduce 30 mol% of Pb excess and the FESEM observations of the films surface do not show presence of pyrochlore in our samples.
Moreover, X-ray diffraction patters confirm presence of PST perovskite phase with no sign or pyrochlore phase. Thus, we are confident that we obtained reasonably good perovskite PST. In addition, the electrical characterisation results, namely relative permittivity measurements and polarization measurements, provide an additional confirmation of presence of PST material.
Modifications in the manuscript:
Line 153-154: “No pyrochlore phase was observed on the micrographs.”
Lines 157-158: “The XRD data show that PST thin films crystallize in perovskite phase without pyrochlore. Samples have two textures”
References:
[1] Zian Kighelman, Films minces relaxeur-ferroelectriques a base de PMN: Elaboration, proprietes dielectriques et electromecaniques, Ph. D. Dissertation, Swiss Federal Institute of Technology Lausanne, 2001
[2] K. Brinkman, Positional order in lead scandium tantalate (PST) as a tol for the investigation of relaxor ferroelectric behavior in thin films, PhD thesis, EPFL Lausanne Switzerland, 2004
- The post-annealing temperature affects crystalline size, composition, and defects in the resultant films. These factors might affect the observed electrical properties of PST films. How to choose the optimal formation temperature of the PST films in this study? Please comment this issue in the revised version.
Indeed, the crystallization temperature effects microstructure, composition, defects and overall quality of the film. In our work we did not perform optimization of deposition conditions of PST, we implemented results of work already published with good results such as Brinkman et al. Our main quality control is FESEM, XRD and then permittivity measurements. We report that our films show
very good electrical and electrocaloric results.
Following modifications have been done in the manuscript, lines 107-108:
“Note that synthesis method and deposition conditions were adapted from [12,18].”
- The resultant film has visible pores in it. Are these visible pores affect reliability of the film for electric device applications?
Indeed, there are visible pores in the cross-section images of the PST films. Nevertheless, this doesn’t prevent outstanding electrical properties of the films, worth sharing with the community. Pores are a problem that will be addressed, they allow room for further improvement.
- The film thickness is 160 nm, but the surface grain size is large (~200 nm). Why?
The film thickness and grain size are two parameters that are not related. In thin films, we often observe columnar microstructure, very typical for highly oriented films, such as our PST. The first deposited layer starts nucleation of grains and puts bases for their size. All additional layers only elongate this grain in the z direction.
Modification in the manuscript line 152:
“The films show a large columnar grain structure”

Reviewer 2 Report
This manuscript reports high electrocaloric effect in lead scandium tantalate thin films with interdigitated electrodes. The results are good. The main originality of this paper is the use of an interdigitated electrode (IDE) geometry with SU8 encapsulation layer enabling the application of very high electric fields up to 1330 kV/cm, which enhanced the electrocaloric effect of PST films significantly. Some revisions are suggested as follows:
- On line 166, is the “The (200) peak outweighs the (220) peak by more than a factor of 15” an error? It may be “The (220) peak outweighs the (200) peak by more than a factor of 15” according to the foregoing sentence.
- On line 197-199, for the sentence “Brinkman reported that relative permittivity of ordered PST thin films is around 6000 and permittivity for disordered films are up to 3000 [12]”, “up” should be revised “down”.
- On 200-204, in the sentence “It is worth noting……for unstrained PST films”, “shift” is referred to Tm not to maximum ε r.
- The Maximum ΔT is not consistent in the manuscript. 14℃ or 15℃?
- On line 243-248, the paragraph “Interestingly enough……The temperature range is stated to be 130 K for PST thin films” does not express clearly. From the sentence, we don’t know whether the so-called temperature range is the electrocaloric temperature variation ΔT or the temperature window of measurement. Is the ΔT is 130K for PST thin films?
- Could you give the actual ratios of I(220)/I(200) according to the XRD data in figure 2 to illustrate more clearly the orientation of films.
- Could you give the results of electrocaloric temperature variation ΔT of different PST thin films on the substrates of c-sapphire and fused silica under lower fields, for example around 774 kV/cm, to compare your film quality with others’ works?
- Can you find the evidence of the stress status of films (tensile, compressive or free) by using their crystal structure change or lattice constant using XRD data? If so, please add it.
Author Response
Dear editor and reviewers,
First, we would like to thank you for your comments, suggestions, and questions. We have implemented changes in response to each point of view and we think that the manuscript has gained in scientific value as a result. The original review comments are shown in black. Answers to those comments are shown in blue and revisions to the manuscript are shown in red font (please see the attached document for colored version).
Reviewer Comments:
Reviewer #2
Open Review
(x) I would not like to sign my review report
( ) I would like to sign my review report
English language and style
( ) Extensive editing of English language and style required
(x) Moderate English changes required
( ) English language and style are fine/minor spell check required
( ) I don't feel qualified to judge about the English language and style
|
Yes |
Can be improved |
Must be improved |
Not applicable |
|
|
Does the introduction provide sufficient background and include all relevant references? |
(x) |
( ) |
( ) |
( ) |
|
Are all the cited references relevant to the research? |
(x) |
( ) |
( ) |
( ) |
|
Is the research design appropriate? |
(x) |
( ) |
( ) |
( ) |
|
Are the methods adequately described? |
(x) |
( ) |
( ) |
( ) |
|
Are the results clearly presented? |
( ) |
(x) |
( ) |
( ) |
|
Are the conclusions supported by the results? |
(x) |
( ) |
( ) |
( ) |
Comments and Suggestions for Authors
This manuscript reports high electrocaloric effect in lead scandium tantalate thin films with interdigitated electrodes. The results are good. The main originality of this paper is the use of an interdigitated electrode (IDE) geometry with SU8 encapsulation layer enabling the application of very high electric fields up to 1330 kV/cm, which enhanced the electrocaloric effect of PST films significantly.
Some revisions are suggested as follows:
- On line 166, is the “The (200) peak outweighs the (220) peak by more than a factor of 15” an error? It may be “The (220) peak outweighs the (200) peak by more than a factor of 15” according to the foregoing sentence.
Thank you for this correction, we changed the error in the sentence.
Corrections in the manuscript lines 160-161:
“The (220) peak outweighs the (200) peak by more than a factor of 15.”
- On line 197-199, for the sentence “Brinkman reported that relative permittivity of ordered PST thin films is around 6000 and permittivity for disordered films are up to 3000 [12]”, “up” should be revised “down”.
Thank you, we corrected this sentence as well.
Corrections in the manuscript lines 190-191:
“permittivity for disordered films are down to 3000”
- On 200-204, in the sentence “It is worth noting……for unstrained PST films”, “shift” is referred to Tm not to maximum ε r.
According to Correira et al. [1] the Tm is defined as the temperature of the peak in εr. Therefore, we are discussing Tm as the maximum of εr. In order to clarify the distinction between the two, we modified the paragraph in the manuscript starting line 193 as follows:
“It is worth noting that the maximum εr is at around 25°C for samples on c-sapphire whereas it is -3°C for samples on FS substrates. Tm is defined as temperature at maximum εr . Therefore, these results are alike the findings in Brinkman et al. [18]. It is reported that the Tm for PST films under tensile or compressive stresses is shifted to lower temperatures than Tm for unstrained PST films. PST on c-sapphire is slightly compressive (-20 MPa [12]), whereas PST on FS is under large tensile stresses [25]. This tendency is confirmed on all the samples, see Supplementary material 5. Thus, these results show that Tm is at around 20°C for c-sapphire and it is shifted to lower temperatures for PST on fused silica. The dielectric loss is gradually decreasing with increasing temperature for PST on fused silica and c-sapphire. “
References:
- Correia, Q. Zhang (eds.), Electrocaloric materials, new generation of coolers, Engineering Materials 34, Springer-Verlag Berlin Heidelberg, 2014, p 72
- The Maximum ΔT is not consistent in the manuscript. 14℃ or 15℃?
Thank you for bringing our attention to this inconsistency. The temperature for maximum ΔT values is 14.5°C. We made this consistent through the manuscript and added corrections to the lines 17, 91, 222, 225, 226, 239, 242, 249, 255.
- On line 243-248, the paragraph “Interestingly enough……The temperature range is stated to be 130 K for PST thin films” does not express clearly. From the sentence, we don’t know whether the so-called temperature range is the electrocaloric temperature variation ΔT or the temperature window of measurement. Is the ΔT is 130K for PST thin films?
Thank you for your comment. The temperature range that we refer to is the working temperature window where the electrocaloric effect was measured and detected. We changed the “temperature range” to “temperature window”. Corrections in the manuscript lines 241 - 246:
“Interestingly enough, for respectively PST on c-sapphire and FS we estimate a ΔT larger than 7 K and 4.5 K for a temperature window between 14.5 to 50°C. This observation confirms the statement of Crossley et al. and Correira et al. that clamping broadens and extends the electrocaloric effect to higher temperatures [1, 19]. The temperature window of measured ΔT for bulk PST is equal to 17 K whereas the window for clamped PST is 70 K [1]. The temperature window is stated to be 130 K for PST thin films [1, 19]. “
- Could you give the actual ratios of I(220)/I(200) according to the XRD data in figure 2 to illustrate more clearly the orientation of films.
The peak fit has been performed with Pymca software from ESRF with the area pseudo-Voigt function and linear function for the background. The peak area is an accurate estimation of the peak intensity. This value was directly fitted:
For C-sapphire:
Area(220) = 750.5 A.U.
Area(200) = 1422.1 A.U.
I(220)/I(200) = 0.53
For fused silica:
Area (220) = 203.7 A.U.
Area (200) = 92.13 A.U.
I(220)/I(200) = 2.21
For both samples, the ratio I(220)/I(200) is lower than 15, thus the main texture is (200).
Please note that XRD pattern in Figure 2 was replaced because of error in peak labelling, Pt(111) replaced with PST (400).
Following modifications have been introduced in the manuscript, lines 161-163:
“The peaks (220) and (200) were fitted with Pymca software and area pseudo-Voigt function with linear function for background. The ratio of integrated areas of (220) and (200) is equal to 2.21 and 0.53 for for PST on c-sapphire and FS respectively.”
- Could you give the results of electrocaloric temperature variation ΔT of different PST thin films on the substrates of c-sapphire and fused silica under lower fields, for example around 774 kV/cm, to compare your film quality with others’ works?
We performed calculations of temperature variation for 233 V corresponding to 774 kV/cm which was the closest value to 774 kV/cm. We added the following figure to the Supplementary material 8:
Figure S6: Estimated temperature variation due to the electrocaloric effect vs temperature of PST on fused silica and c-sapphire under 1330 kV/cm and 776 kV/cm.
We added lines 234 to 236 to the main manuscript:
“For comparison, calculations of ΔT under 776 kV/cm are reported in Figure S6 in Supplementary material 8. Under 776 kV/cm, the highest ΔT is equal to 11.5 K and 9.1 K for c-sapphire and FS respectively. These values are 1.3 et 1.7 times larger than reported by Correia et al.”
- Can you find the evidence of the stress status of films (tensile, compressive or free) by using their crystal structure change or lattice constant using XRD data? If so, please add it.
In order to assess the strain and for convenience in peak fitting, we took the PST (400) peak for consideration for strain assessment. According to PDF 01-074-2635, the 2θ angle for (400) peak is located at 44.48° in unstrained powder. The PST (400) peak in film on c-sapphire is located at 2θ equal to 44.48° also. Same 2θ angle means that lattice parameter, or d-spacing, is unchanged. Therefore, one can say that PST on c-sapphire is unstrained, and unstressed. This is expected because the thermal expansion coefficients of PST and c-sapphire are close, 6.5 10-6 K-1 and 6.7 10-6 K-1 respectively.
In the PST film on fused silica, the (400) peak is located at 2θ equal to 44.56°. Larger 2θ corresponds to smaller d-spacing in the probing direction. The probing direction of XRD is the out-of-plane direction. Therefore, there is smaller d-spacing in the out-of-plane direction. Smaller d-spacing in the out-of-plane direction correlates to larger d-spacing in the in-plane direction, thus to the tensile strain of the thin film. This is expected. PST film on fused silica is expected to be under tensile strain, because of the thermal expansion coefficient mismatch between PST and fused silica, 6.5 10-6 K-1 and 0.48 10-6 K-1 respectively [1].
Calculation of PST stress from thermal expansion coefficient:
Where is stress, is Young’s modulus of the film, is Poisson’s ratio of the film, is thermal expansion coefficient of the film and is the thermal expansion coefficient of the substrate, and is the temperature variation. Taking equal to 1 1011 Nm-2 , equal to 0.3, as mentioned above, and equal to 875 K, the expected tensile stress in PST film on fused silica is 752.5 MPa.
The latter paragraph is added to Supplementary material 5.
Modifications in the manuscript lines 197-198:
“whereas PST on FS is under large tensile stresses (+752 MPa see Supplementary material, [25]).
Reference:
- Swift glass. Available online : https://www.swiftglass.com/blog/material-month-fused-silica/ (26/04/2022)

Reviewer 3 Report
- Introduction: “These achievements have been possible due to improved ordering S of PST”. Please define “S” otherwise this would be confused with S in ΔS which has been defined before.
- The introduction has more than enough general information but no discussion or statement on the goal or the issue addressed in the work is found. More specifically, the author should focus on the “interdigitated electrodes”. What is the current issue without it? What could be the benefits (as the researchers aimed before proceeding with the work)? Please revise the introduction.
- Recently, ECE researchers are talking about reliability (on electric field cycling and thermal cycling). Where does this device stand on this aspect? Or at least what can be expected if “interdigitated electrodes” are adopted.
- Which temperature unit was used in equations 1 and 2 for calculation? What about figure 6? Please use the proper scale in figure 5 so that the variation is visible.
- Please re-check the calculation. It is surprising that ΔT is high even though the dP/dT is very small. Please explain.
- What is the contribution of the “interdigitated electrodes”? This has never been discussed in the entire results and discussion. I could not find a single analysis or comparison of this with related results.
- The supplementary materials are just the copy of the figures presented in the manuscript with a few additional things. Please keep only those that are not presented in the manuscript.
- Please show the entropy change graph (ΔS vs T).
- There is a contradiction in the temperature dependence of ECE presented in this work with the work reported by Correira et al. Please explain.
Author Response
Dear editor and reviewers,
First, we would like to thank you for your comments, suggestions, and questions. We have implemented changes in response to each point of view and we think that the manuscript has gained in scientific value as a result. The original review comments are shown in black. Answers to those comments are shown in blue and revisions to the manuscript are shown in red font. Please see the attached document.

Round 2
Reviewer 1 Report
The authors have revised the manuscript according to review comments point by point. The scientific quality of the submitted work has improved. The revised version is acceptable for publication.
Reviewer 3 Report
Improved.